# Number of antenatal care visits and associated factors among reproductive age women in Sub-Saharan Africa using recent demographic and health survey data from 2008–2019: A multilevel negative binomial regression model

**Fetene Getnet Gebeyehu**[1], **Bisrat Misganaw Geremew**[2], **Aysheshim Kassahun Belew**[3], **Melkamu Aderajew Zemene**[4] *

1 Department of Epidemiology, Gambella Regional Health Bureau, Gambella, Ethiopia, 2 Department of Epidemiology and Biostatistics, Institute of Public Health, College of Medicine and Health Sciences, University of Gondar, Gondar, Ethiopia, 3 Department of Human Nutrition, Institute of Public Health, College of Medicine and Health Sciences, University of Gondar, Gondar, Ethiopia, 4 Department of Public Health, College of Health Sciences, Debre Tabor University, Debre Tabor, Ethiopia

* melmahman3m@gmail.com

## Abstract

### Background

Antenatal care is one of the best strategies for maternal and neonatal mortality reduction. There is a paucity of evidence on the mean number of ANC visits and associated factors in Sub-Saharan Africa (SSA). This study aimed to investigate the mean number of ANC visits and associated factors among reproductive-age women in Sub-Saharan Africa using the Demographic and Health Survey conducted from 2008 to 2019.

### Method

A total of 256,425 weighted numbers of women who gave birth five years before the survey were included. We used STATA version 14 for data management and analysis. A multilevel negative binomial regression model was fitted. Finally, the Adjusted Incident Rate Ratio (AIRR) with its 95% CI confidence interval was reported. Statistical significance was declared at P-value < 0.05.

### Results

The mean number of ANC visits among women who gave birth five years before the survey in SSA was 3.83 (95% CI = 3.82, 3.84) Individual-level factors such as being aged 36–49 years (AIRR = 1.20, 95% CI = 1.18,1.21), having secondary education &above (AIRR = 1.44, 95% CI = 1.42, 1.45), having rich wealth status (AIRR = 1.08, 95% CI = 1.07, 1.09), media exposure (AIRR = 1.10, 95% CI = 1.09,1.11), and grand multiparity (AIRR = 0.90,

**Data Availability Statement:** The dataset is available from the DHS program official database www.measuredhs.com.

**Funding:** The authors received no specific funding for this work.

**Competing interests:** The authors have declared that no competing interests exist.

95% CI = 0.89, 0.91) were significantly associated with the number of ANC visits. Furthermore, rural residence (AIRR = 0.90, 95% CI = 0.89, 0.91), Western SSA region (AIRR = 1.19, 95% CI = 1.18, 1.20) and being from a middle-income country (AIRR = 1.09, 95% CI = 1.08, 1.10) were community-level factors that had a significant association with the number of ANC visits.

## Conclusion

The mean number of ANC visits in SSA approximates the minimum recommended number of ANC visits by the World Health Organization. Women's educational status, women's age, media exposure, parity, planned pregnancy, wealth status, residence, country's income, and region of SSA had a significant association with the frequency of ANC visits. This study suggests that addressing geographical disparities and socio-economic inequalities will help to alleviate the reduced utilization of ANC services.

## Background

Antenatal care (ANC) is care provided by skilled healthcare professionals to pregnant women to ensure the best health conditions for both the mother and fetus during pregnancy [1]. ANC decreases maternal and perinatal morbidity and mortality [2]. The ANC service includes birth preparedness, advice on danger signs of pregnancy, counseling on optimal nutrition, prevention, identification and treatment of obstetric complications, and advice on options for family planning [3].

Maternal death is defined as the death of a woman while pregnant or within 42 days of the pregnancy's termination from any cause related to or aggravated by the pregnancy or its management, but not from accidental or incidental causes [4]. Child and maternal mortality continued to be major public health concerns in developing countries. Yearly, 527,000 women in low-income countries die from pregnancy-related complications, and nearly 4 million neonates die in their first of which 98% are from developing countries [5].

According to the World Health Organization (WHO) sustainable development goal (SDG) 3, countries should reduce child mortality to less than 25 deaths per 1,000 live births and maternal mortality to less than 70 per 100,000 live births by the year 2030 [6]. Although 121 countries had met the target on under-5 mortality, progress will need to accelerate in 53 countries, two-thirds of which are in sub-Saharan Africa [7]. Studies showed that utilization of at least one antenatal care visit by a skilled provider during pregnancy decreases the risk of neonatal mortality by 39% in sub-Saharan African countries. Thus, to accelerate progress toward the reduction of neonatal mortality, all pregnant women should receive antenatal care [8].

Globally, 87% of pregnant women received antenatal care visits at least once, and only 60% received the minimum required number of antenatal care visits [3]. In Sub-Saharan Africa, 49–53% of women received the minimum recommended number of ANC visits 35% accessed at least one visit and 13% of women had no antenatal care visits [9].

Due to the high burden of maternal and child mortality, WHO adopted the focused antenatal care model by the year 2002, which recommends a minimum of four antenatal care visits by a pregnant woman [10]. By the year 2016, at the start of the Sustainable Development Goal era, maternal and child mortality was unacceptably high and the world health organization

changed the focused ANC model to an essential core package of routine ANC and the number of visits increased to eight [1].

Previous studies affirmed that Place of residence [11, 12], wealth status [13, 14], educational status [15, 16], distance from a health facility [17, 18], quality of health service [19, 20], and cost of service [21] were factors associated with the number of antenatal care visit. Barriers to quality maternal health care must be recognized and addressed at all levels of the health system to enhance maternal health [22].

Though there are studies on the prevalence of ANC and associated factors in different African countries, we found limited evidence on the mean number of ANC visits and associated factors in SSA using the count model. Since most of the studies were conducted using binary logistic regression, information was lost while categorizing women's different numbers of visits in the same group. Most of the studies focused on factors associated with the timing of the first ANC initiation. There is a paucity of information on the average number of ANC visits and associated factors among women in SSA.

The count model has the added benefit of estimating the mean number of ANC visits and associated factors, and information loss may not be an issue. Therefore, this study aimed to investigate the mean number of antenatal care visits and associated factors among reproductive-age women in SSA based on the 2008 to 2019 Demographic and Health Survey data.

## Methods and materials

### Study design, area, and period

Secondary data analysis was conducted based on the recent Demographic and Health Survey (DHS) data of 35 Sub-Saharan African countries from 2008 to 2019. According to the United Nations geo scheme classification, the continent of Africa is commonly divided into five regions [23]. Sub-Saharan Africa contains four of these regions namely Eastern Africa, Central Africa, Western Africa, and Southern Africa. This study was conducted in these regions using their recent DHS datasets.

### Source and study population

The source population was all reproductive-age women who gave birth five years before each respective survey in Sub-Saharan Africa, whereas those in the selected Enumeration Areas (EAs) were the study population. The sample size was determined from the kids to recode file "KR file" from the standard DHS dataset of Sub-Saharan African countries with at least one survey from 2008 to 2019. DHS uses a two-stage stratified cluster sampling technique. In the first stage, a sample of EAs was selected independently from each stratum with proportional allocation stratified by residence (urban & rural). In the second stage, from the selected EAs, households were taken by systematic sampling technique [24]. The final sample size was 257,924 (weighted sample 256, 425) women.

### Variables and data collection procedure

The outcome variable was the number of antenatal care visits. The independent variables for this study were from two sources such as individual and community-level factors. The individual-level factors include; socio-economic and demographic-related factors, pregnancy-related factors, and behavioral-related factors. Community-level factors include; community-level media exposure, community-level women's education, place of residence, sub-regions within SSA, and year of the survey. The data were accessed and downloaded from the webpage of the international Demographic and Health Survey.

## Operational definitions

**The number of antenatal care visits.**   Non-negative integer for which this study aims to identify a proper count regression model.

**Media exposure.**   It was obtained by aggregating women's exposure to television, radio, and newspaper and if women had been exposed at least once a week it was coded as "1" for yes, and if a woman had not been exposed at least once a week it was coded as "0" for no.

**Wealth status.**   Is a composite measure of a household's cumulative living standard divided into 5 quantiles using the wealth quantile data derived from principal component analysis. Finally, it was coded as "0" for the poor, "1" for the middle, and "2" for the rich.

**Community-level media exposure.**   Was obtained by aggregating the individual level media exposure at the cluster level by using the proportion of women who had media exposure and it was coded as "0" for low (communities in which < 50% of women had media exposure), "1" for high community-level media exposure. This community-level media exposure shows the overall media exposure in the community.

**Community-women education.**   It was obtained by aggregating the individual-level women's education into clusters by using the proportion of women who had attended at least primary education. It was coded as "0" for low (communities in which < 50% of women had at least primary education), and "1" for high community-Community-level.

**Year of the survey.**   The period when the survey was conducted.

## Data analysis

The data was cleaned, coded, and extracted using MS excel and STATA version 14 software. Variables with missed values greater than 5% / not applicable were dropped. During data cleaning and coding, the same STATA command was applied for each country. After coding and cleaning the dataset for each country independently, all country dataset was appended to one dataset for further analysis. Sample weighting was done for each country before appending it to one dataset.

## Poisson regression model

The standard Poisson regression model was the first model considered while working with count data. It is a technique used to describe count data as a function of a set of independent variables and which assumes the observations should be independent over time and the mean and the variance of the dependent variable should be equal [25].

However, the assumption of the Poisson regression failed (the sample variance exceeds the sample mean), in the case of overdispersion. In such a case the negative binomial regression model that includes an unobserved specific effect (random term or error term) for the parameter was the preferred one to handle the overdispersion. A Likelihood Ratio(LR) test for the parameter $\alpha$ (p-value < 0.001) was used for the over-dispersion parameter, in the negative binomial (NB) specification against the Poisson model specification [26].

## Multilevel model building

Since DHS data has hierarchical nature different measures of variation (intra-class correlation coefficient (ICC), and Median Incident Rate Ratio (MIRR)) were calculated to detect any clustering effect. Finally, four multilevel count models were fitted.

First, a "**null**" model (**model 1**), which only includes a random intercept and allows us to detect the existence of a possible contextual dimension for a phenomenon was estimated and provided evidence to assess random effect using the Intraclass Correlation Coefficient (ICC).

Then the individual characteristics were included in the model (**model 2**) to investigate the extent to which the overall difference in the number of ANC visits was explained by the individual variation. Next, community-level variables were added to the model (**model 3**) to investigate whether this contextual phenomenon was conditioned by community-level characteristics. Finally, both individual and community-level characteristics were added to the model (**model 4**) at the same time as the number of ANC visits.

## Parameter estimation method

The fixed effects (a measure of association) were used to estimate the association between explanatory variables and the number of ANC visits at both individual and community levels. Factors with a p-value $\leq 0.25$ were selected as candidates for the final model. The crude Incident Rate Ratio (CIRR) and the Adjusted Incident Rate Ratio (AIRR) were assessed and finally Adjusted Rate Ratio (AIRR) was presented. Associations between dependent and independent variables were assessed and their strength was presented using adjusted rate ratios and 95% confidence intervals with a p-value of $<0.05$.

Random effects (a measure of variation) were estimated by the intraclass correlation (ICC). ICC was used to assess the cluster variation. As well the percent change of variance (PCV) which measures the total variation attributed to individual-level and cluster-level factors was calculated. The ICC was calculated by using the formula [27]

$$\text{ICC} = \text{the variance of the null model} \,/\, (\text{variance of the null model} \,+\, \frac{\pi^2}{2})$$

The MIRR is defined as the median value of the incident rate ratio between the area at the highest risk (reduced ANC visit) and the area at the lowest risk (increased ANC visit) when randomly picking out two clusters. The MIRR was calculated by the formula, [27]

$$\text{MIRR} = e^{0.95\sqrt{\delta^2}}, \text{ where } \delta^2 = \text{cluster level variance.}$$

The variance inflation factor (VIF) was assessed to check for multi-linearity and the mean VIF for the fitted model was 1.86. Finally, a model comparison was done using the deviance test, and the model with the lowest deviance was selected as the best-fit model.

## Ethical consideration

The waiver of written informed consent was approved by the University of Gondar Institutional Review Board (IRB). As well, after online requesting and explaining the objective of this study to DHS Program/ICF International Inc., a letter of permission was waived from the International Review Board of Demographic and Health Surveys (DHS) program data archivists to download the dataset for this study. The data was not shared or passed on to other researchers.

## Results

### Socio-demographic characteristics of respondents

A total of 257,924 reproductive-age women who gave birth within five years before the survey were included. The median age of women was 28 years with Inter Quartile Range (IQR) of 23–34 years. Nearly three fourth (72.3%) of the study participants were between the age of 20–35 years. More than one-third (38.43%)of the participants had no formal education. Nearly half (45.71%) of the respondents were from poor and poorest households, and about two-thirds (66.85%) were from rural areas. More than three fourth (77%) of the respondents were from

the eastern and western regions of sub-Saharan Africa. And approximately one-third (28%) of the pregnancies were unwanted (Table 1).

## Number of ANC visits during pregnancy

The mean number of ANC visits in SSA was 3.83 (95% CI = 3.82, 3.84) with the minimum average number of visits among Chadian women (2.29, 95% CI = 2.26, 2.34) and maximum average number of visits among Ghanan women (6.44, 95% CI = 6.35, 6.53). There was also a difference in the mean numbers of ANC visit among regions of SSA with 3.55 (95% CI = 3.53, 3.58) for women from the Central region, 3.66 (95% CI = 3.65, 3.67) for women from the Eastern region, 4.02 (95% CI = 4.00, 4.04) for women from Western region and 4.99 (95% CI = 4.93, 5.06) for women from Southern region.

**Table 1. Weighted socio-demographic and economic characteristics of respondents DHS 2008–2019 (N = 256,425).**

| Variables | Frequency | Percentage |
|---|---|---|
| Maternal age | | |
| 15–19 | 19,543 | 7.62 |
| 20–35 | 186,738 | 72.82 |
| 36–49 | 50,143 | 19.55 |
| Maternal education | | |
| No education | 95,643 | 37.30 |
| Primary | 87,083 | 33.96 |
| Secondary& above | 73,699 | 28.74 |
| Wealth index | | |
| Poor | 108,661 | 42.38 |
| Middle | 51,287 | 20.00 |
| Rich | 96,476 | 37.62 |
| Media usage | | |
| Yes | 168,980 | 66.00 |
| No | 87,040 | 34.00 |
| Parity | | |
| Primi | 54,319 | 21.18 |
| Multi | 147,2934 | 57.44 |
| Grand | 54,812 | 21.38 |
| Residence | | |
| Rural | 171,412 | 66.85 |
| Urban | 85,013 | 33.15 |
| Region of sub-Saharan Africa | | |
| Western | 100,111 | 39.04 |
| Eastern | 98,663 | 38.48 |
| Central | 48,221 | 18.81 |
| Southern | 9,429 | 3.68 |
| Country GDP | | |
| Low | 167,900 | 65.48 |
| Middle | 88,525 | 34.52 |
| Pregnancy | | |
| planned | 180,144 | 72.41 |
| Unplanned | 68,629 | 27.59 |

## Multilevel negative binomial regression analysis

### Random effect and model comparison results

Although we considered fitting different count models, we preferred the negative binomial regression model to the standard Poisson regression model as the sample variance (6.6) exceeds the sample mean (3.83) and the Likelihood Ratio (LR) test became significant (p-value <0.001). The data were also checked for excess zeros to determine if there was a possibility to choose models that are suitable for data with an inflated number of zeros (ZIP, ZINB, HP, and HNB). Different measures of variations were calculated to determine whether there is a clustering effect or not. First, the intra-class correlation was calculated and it affirmed that the total variability in the number of ANC visits explained by the cluster variation was only 1.1% (Table 2). This indicated that the cluster variation was not large to use a multilevel model instead of the standard negative binomial regression model, but it is not always true [28].

Furthermore, we could use a likelihood ratio test to compare the likely hood of the fitted multilevel model with the standard model [29]. The p-value associated with the chi-squared statistics was less than 0.001, hence we would reject the null hypothesis of no between-cluster variation in the rate of occurrence of the outcome (number of ANC visits). In addition, the median incident rate ratio (MIRR) allows us to determine the median relative change in the rate of occurrence of the outcome (number of ANC visits) between an individual in a cluster with a high rate of outcome (ANC visit) and an individual from a cluster with a low rate of outcome (ANC visit) was calculated. The MIRR for the null model became significant (MIRR = 1.20, 95% CI = 1.18,1,21) (Table 2), which tells us doing a multilevel model was preferred over the standard model [29].

The percent change of variance (PCV) which measures the total variation attributed to individual-level and cluster-level factors was calculated. The deviance test was used to select the best-fitted model and its values are decreasing across model 1 to model 4. Therefore model 4 was the model with the lowest deviance test value and it became the preferred model. The Percent change variation in the final model showed that about 63.6% of the total variability in the number of antenatal care visits was explained by the full model.

### Fixed effect results; multilevel negative binomial regression analysis

Maternal age, maternal education, wealth status, media usage, and parity are individual-level factors that had a significant association with the number of ANC visits in the final model. The

**Table 2. Parameters and model fitness statistics for multilevel negative binomial regression analysis.**

| Parameters | Null model | Model 2 | Model 3 | Model 4 |
|---|---|---|---|---|
| Cluster level variance (SE) | 0.036 (0.001) | 0.027(0.0008) | 0.023 (0.0007) | 0.022 (0.0007) |
| ICC | 0.18% | | | |
| MIRR | 1.20 (95% CI = 1.18, 1.21) | 1.17(95% CI = 1.16, 1.17) | 1.15(95% CI = 1.15, 1.16) | 1.15(95% CI = 1.15, 1.16) |
| PCV | Reff. | 33.3% | 56.5% | 63.6% |
| Model fitness | | | | |
| Deviance | 1,169,652 | 1,157,281 | 1,150,418 | 1,098,785 |

ICC = intra-class correlation coefficient; MIRR = median incident rate ratio; PCV = proportional change in variance.

Model I is the null model, a baseline model without any independent variable.

Model II is adjusted for individual-level factors.

Model III is adjusted for community-level factors.

Model IV is the full model adjusted for both individual and community-level factors.

frequency of ANC visits increased by 8% (AIRR = 1.08, CI = 1.07, 1.10) and 18% (AIRR = 1.18, 95% CI = 1.16, 1.20) for women aged 20–35 and above 35 as compared to women of 15–19 years of age, respectively. Primary education and secondary & above had increased the frequency of ANC visits by 26% (AIRR = 1.26, 95% CI = 1.25, 1.27) and 44% (AIRR = 1.44, 95% CI = 1.42, 1.45) compared with women with no education, respectively. Middle and rich wealth status had increased the frequency of ANC visits by 5% (AIRR = 1.05, 95% CI = 1.04, 1.06) and 8% (AIRR = 1.08, 95% CI = 1.07, 1.09) respectively as compared to women of poor wealth status. Women's media exposure had increased the frequency of ANC visits by 10% (AIRR = 1.10, 95% CI = 1.09, 1.11) when compared to women with no media exposure. Whereas multiparity and grand multiparity decreased the frequency of ANC visits by 2% (AIRR = 0.98, 95% CI = 0.97, 0.99) and 10% (AIRR = 0.90, 95% CI = 0.89, 0.91) respectively when compared with primiparity. Women with planned pregnancies had 3 percentage points (AIRR = 0.97, 95% CI = 0.96, 0.98) reduced frequency of ANC visits when compared with their counterparts.

Furthermore, community-level factors were significantly associated with the frequency of ANC visits. Women from the rural residence had 10% (AIRR = 0.90, 95% CI = 0.89–0.91) lower ANC visits than their counterparts. Women from Eastern, Southern, and Western SSA regions had 3% (AIRR = 1.03, 95% CI = 1.02, 1.04), 11% (AIRR = 1.11, 95% CI = 1.09, 1.13), and 17% (AIRR = 1.17, 95% CI = 1.16, 1.18) number of ANC visit respectively when compared to women from central SSA region. Women from the community with a high level of media exposure and high level of women's education had 3% (AIRR = 1.03, 95% CI = 1.02, 1.04) and 2% (AIRR = 1.02, 95% CI = 1.01–1.03) increment with a frequency of ANC visit respectively compared with those from a lower level. Women from middle-income countries had 10% (AIRR = 1.10, 95% CI = 1.09, 1.11) of a greater number of ANC visits compared to their counterparts. In addition, women surveyed from 2012–2015 and 2016–2019 had 5% (AIRR = 1.05, 0.95% CI = 1.04, 1.06) and 7% (AIRR = 1.07, CI = 1.06, 1.08) a greater number of ANC visits when compared with women surveyed from 2008–2011 (Table 3).

## Discussion

Antenatal care is an indicator to measure the efficiency of maternal care utilization. It helps in preventing adverse pregnancy outcomes when provided early in the pregnancy and continued through delivery. Identification of problems in pregnancy results in early referrals for women with complications. So this study focused on the mean number of ANC visits and determinants of the number of Antenatal care visits in SSA using the multilevel negative binomial analysis to estimate individual and community-level factors.

This study revealed that although the mean number of ANC visits in SSA approximates the minimum recommended number of ANC visits by the World Health Organization (WHO) [1], there were still disparities from region to region regarding the number of visits. This might have resulted from inequalities in the accessibility of maternal health services, poor/absence of transportation, inequality in the number of health care providers, and disparities in access to education [30, 31]. In addition, this disparity might be due to the difference in the country's policy and program implementation regarding maternal health service delivery, women's education, and the role of women in household wealth status [32]. The other possible reason might be the difference in the implementation of different maternal and child health programs among regions of SSA.

In this study individual and community-level factors are responsible for approximately 64% of differences in the number of ANC visits during pregnancy in Sub-Saharan Africa. In the current study women aged 36–49 and 20–35 years were eighteen and eight percentage points more likely to visit health institutions for ANC service when compared to women aged

**Table 3. Multilevel negative binomial regression analysis of individual and community level factors associated with the number of antenatal care visits in sub-Saharan Africa, DHS 2008–2019 (N = 256,425).**

| Variables | category | Model_2 | Model_3 | Model_4 |
|---|---|---|---|---|
| | | AIRR[95% CI] | AIRR[95% CI] | AIRR[95% CI] |
| Maternal age | 15–19 | 1.00 | ________ | 1.00 |
| | 20–35 | 1.09 [1.08, 1.10] *** | ________ | 1.08 [1.07, 1.10]*** |
| | 36–49 | 1.20 [1.18, 1.21] *** | ________ | 1.18 [1.16, 1.20]*** |
| Maternal education | No educ. | 1.00 | ________ | 1.00 |
| | Primary | 1.20 [1.19, 1.21] *** | ________ | 1.26 [1.25, 1.27] *** |
| | Secondary & above | 1.46 [1.45, 1.47] *** | ________ | 1.44 [1.42, 1.45] *** |
| Wealth index | Poor | 1.00 | ________ | 1.00 |
| | Middle | 1.06 [1.05, 1.07] *** | ________ | 1.05 [1.04, 1.06]*** |
| | Rich | 1.13 [1.12, 1.14] *** | ________ | 1.08 [1.07, 1.09] *** |
| Parity | Primi | 1.00 | ________ | 1.00 |
| | Multi | 0.97 [0.96, 0.98] *** | ________ | 0.98 [0.97, 0.99]*** |
| | Grand | 0.88 [0.87, 0.90] *** | ________ | 0.90 [0.89, 0.91] *** |
| Media usage | No | 1.00 | ________ | 1.00 |
| | Yes | 1.14 [1.13, 1.15] *** | ________ | 1.10 [1.09, 1.11] *** |
| Pregnancy | planned | 1.00 | | 1.00 |
| | Unplanned | 0.96 [0.95, 0.97] *** | | 0.97 [0.96, 0.98] *** |
| Residence | Urban | ______ | 1.00 | 1.00 |
| | Rural | ________ | 0.76 [0.75, 0.77] *** | 0.90 [0.89, 0.91] *** |
| Region of SSA | Central | ______ | 1.00 | 1.00 |
| | Southern | ________ | 1.24 [1.22, 1.26] *** | 1.11 [1.09, 1.13] *** |
| | Eastern | ________ | 1.08 [1.07, 1.09] *** | 1.03 [1.02, 1.04] *** |
| | Western | ________ | 1.13 [1.12, 1.14] *** | 1.17 [1.16, 1.18] *** |
| Country income | Low | ________ | 1.00 | 1.00 |
| | Middle | ________ | 1.14 [1.13, 1.15] *** | 1.10 [1.08, 1.11] *** |
| DHS survey year | 2008–2011 | ________ | 1.00 | 1.00 |
| | 2012–2015 | | 1.06 [1.05, 1.07] *** | 1.05 [1.04, 1.06] *** |
| | 2016–2019 | | 1.08 [1.07, 1.09] *** | 1.07 [1.06, 1.08] *** |
| Community media usage | Low | ________ | 1.00 | 1.00 |
| | High | ________ | 1.06 [1.05, 1.07] *** | 1.03 [1.02, 1.04] *** |
| Community women's educ. | Low | ________ | 1.00 | 1.00 |
| | High | ________ | 1.09 [1.08, 1.10] *** | [1.01, 1.03] *** |

* = p-value < 0.05,

** = p-value < 0.01,

*** = p-value < 0.001, AIRR = adjusted incidence rate ratio.

Model I is the null model, a baseline model without any independent variable.

Model II is adjusted for individual-level factors.

Model III is adjusted for community-level factors.

Model IV is the full model adjusted for both individual and community-level factors.

15–19 years. The finding supports other studies done previously in different countries [33, 34] that showed a positive association between ANC visits and increased age of women. This might be due to birth-related complications and poor health conditions as age advances which trigger the women to demand more visits. In addition, it indicates that young women (15–19 years) probably lack experience in pregnancy care compared to older women [9].

Our finding also showed that women who had attained primary and secondary and above education were 26 and 44 percentage points more likely to have frequent ANC visits when compared with women with no education respectively. This finding is similar to other studies [16, 35–38] conducted before in SSA as well as in other countries. This might be a result of the improvement in health literacy as the educational level increase [39].

Another factor that had a significant relationship with the frequency of ANC visits was whether the pregnancy was planned or not. The study indicated that unwanted pregnancy was 3 percentage points more likely to have a reduced frequency of ANC visits. The finding was consistent with the findings of other studies done before [38, 40–42]. It is obvious that if the pregnancy was wanted women's willingness to get health services would increase and there might be early detection of pregnancy, which in turn leads to early booking for ANC, as a result, the frequency of ANC visits would increase.

Furthermore, this study showed that the wealth status of the woman had a positive and significant effect on the frequency of ANC visits. Women from the middle and rich wealth quintiles were more likely to have a frequent number of ANC visits than women from the poor wealth quintile. This finding was consistent with the findings in other studies conducted in different countries [13, 14, 17, 43]. This indicates that wealth status is an important variable that can influence the frequency of ANC visits. This could be because poor women can not afford transportation fees to a health facility for ANC service utilization. Besides, it can also affect the utilization of ANC services indirectly due to a lack of media exposure and access to education when compared to women with a high wealth status [44, 45].

In this study, women's exposure to mass media has a significant effect on the number of ANC visits. The result showed that women who had media exposure were more likely to have frequent ANC visits than their counterparts. This finding is consistent with previous research [9, 11, 17]. The possible explanation for this finding is that providing women with adequate information about maternal health services increases their utilization of such services. Parity is another factor that had a significant relationship with the frequency of ANC visits. According to our findings, primipara women were more likely to have frequent ANC visits than those with multiparity. In other words, the greater the women's parity, the less likely they were to have frequent visits. This finding is supported by other previous studies [46–48]. This might be a result of increased confidence from previous birth experiences [37, 49–51].

Women from rural residences were less likely to have frequent ANC visits. The result was consistent with the findings of studies conducted in different countries [44, 45, 52]. The possible explanation for this is the lack of health facilities in rural areas as compared to urban settings. Furthermore, rural women have no access to health-related information.

Moreover, the geographical region of SSA was found to be a significant factor that could affect the frequency of ANC visits. This study was in line with previously conducted studies [11, 16, 17]. This could be due to disparities in access to health facilities, transportation, and socioeconomic differences.

Our study revealed that women in middle-income countries were more likely to have frequent ANC visits than women in low-income countries. Although no similar study has been conducted, the possible explanations will be improved access to health services, and transportation as the income is higher. Regarding the DHS survey year, women surveyed from 2012–2015 and 2016–2019 were 5 and 7 percentage points more likely to have a greater number of visits than women surveyed from 2008–2011. This might be due to the advancement in the accessibility of health service infrastructures, and the increment in the number of health professionals from time to time.

The current study found that women from high-education communities were more likely to visit healthcare facilities during pregnancy than women from low-education communities.

This finding is supported by another study [34]. This can be explained by herd health literacy where health-related information will be easily accessible in the community.

Women with a high level of media exposure are more likely to have ANC visits than women with a low level of media exposure. The explanation could be that access to the media is an enabling factor for ANC service utilization.

## Strength and limitations

This study used large population-based data with a large sample size, which is representative of 35 sub–Saharan African countries. Furthermore, a count data analysis (multilevel Negative Binomial regression analysis) was applied which enabled us to model the effects of each determinant on the frequency of ANC visit efficiently. The novelty of this paper lies in the fact that we have modeled the determinants of the number of antenatal care services in SSA using the most recent DHS data for each country. One significant point of departure of this study is that some countries had no recent DHS data and data from some other countries are not publicly available. Despite the cross-sectional nature of the DHS data, reports of this finding are explained by the incidence rate ratio.

## Conclusion

The mean number of ANC visits in SSA approximates the minimum recommended number of ANC visits by the WHO. Maternal education, maternal age above 20 years, media exposure, rich wealth status, high level of community education, high level of media exposure, country GDP, and being from the western and southern regions of SSA increased the frequency of ANC visits. On the other hand, rural residence, multiparity, and unplanned pregnancy negatively affected the frequency of ANC visits. Therefore, this study suggests that addressing geographical disparities and socio-economic inequalities will help alleviate the reduced utilization of ANC services.

## Acknowledgments

The authors would like to thank the MEASURE DHS program for the on-request open access to its dataset.

## Author Contributions

**Conceptualization:** Fetene Getnet Gebeyehu, Bisrat Misganaw Geremew, Aysheshim Kassahun Belew, Melkamu Aderajew Zemene.

**Data curation:** Fetene Getnet Gebeyehu, Bisrat Misganaw Geremew, Aysheshim Kassahun Belew, Melkamu Aderajew Zemene.

**Formal analysis:** Fetene Getnet Gebeyehu, Bisrat Misganaw Geremew, Aysheshim Kassahun Belew, Melkamu Aderajew Zemene.

**Funding acquisition:** Fetene Getnet Gebeyehu, Bisrat Misganaw Geremew, Aysheshim Kassahun Belew, Melkamu Aderajew Zemene.

**Investigation:** Fetene Getnet Gebeyehu, Bisrat Misganaw Geremew, Aysheshim Kassahun Belew, Melkamu Aderajew Zemene.

**Methodology:** Fetene Getnet Gebeyehu, Bisrat Misganaw Geremew, Aysheshim Kassahun Belew, Melkamu Aderajew Zemene.

**Project administration:** Fetene Getnet Gebeyehu, Bisrat Misganaw Geremew, Aysheshim Kassahun Belew, Melkamu Aderajew Zemene.

**Resources:** Fetene Getnet Gebeyehu, Bisrat Misganaw Geremew, Aysheshim Kassahun Belew, Melkamu Aderajew Zemene.

**Software:** Fetene Getnet Gebeyehu, Bisrat Misganaw Geremew, Aysheshim Kassahun Belew, Melkamu Aderajew Zemene.

**Supervision:** Fetene Getnet Gebeyehu, Bisrat Misganaw Geremew, Aysheshim Kassahun Belew, Melkamu Aderajew Zemene.

**Validation:** Fetene Getnet Gebeyehu, Bisrat Misganaw Geremew, Aysheshim Kassahun Belew, Melkamu Aderajew Zemene.

**Visualization:** Fetene Getnet Gebeyehu, Bisrat Misganaw Geremew, Aysheshim Kassahun Belew, Melkamu Aderajew Zemene.

**Writing – original draft:** Fetene Getnet Gebeyehu, Bisrat Misganaw Geremew, Aysheshim Kassahun Belew, Melkamu Aderajew Zemene.

**Writing – review & editing:** Fetene Getnet Gebeyehu, Bisrat Misganaw Geremew, Aysheshim Kassahun Belew, Melkamu Aderajew Zemene.

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
