## [Decision Letter · Decision Letter 0]

8 Sep 2022

PGPH-D-22-00897

Number of antenatal care visits and associated factors among reproductive age women in Sub-Saharan Africa using recent demographic and health survey data from 2008-2019: a multilevel negative binomial regression model

Dear Dr.Zemene,

Thank you for submitting your manuscript to PLOS Global Public Health. After careful consideration, we feel that it has merit but does not fully meet PLOS Global Public Health’s publication criteria as it currently stands. Therefore, we invite you to submit a revised version of the manuscript that addresses the points raised during the review process.

We look forward to receiving your revised manuscript.

Kind regards,

Jitendra K. Singh, PhD

Academic Editor

Journal Requirements:

Additional Editor Comments (if provided):

Reviewers' comments:

Reviewer's Responses to Questions

**Comments to the Author**

1. Does this manuscript meet PLOS Global Public Health’s publication criteria? Is the manuscript technically sound, and do the data support the conclusions? The manuscript must describe methodologically and ethically rigorous research with conclusions that are appropriately drawn based on the data presented.

Reviewer #1: Partly

Reviewer #2: No

Reviewer #3: Partly

2. Has the statistical analysis been performed appropriately and rigorously?

Reviewer #1: I don't know

Reviewer #2: No

Reviewer #3: Yes

3. Have the authors made all data underlying the findings in their manuscript fully available (please refer to the Data Availability Statement at the start of the manuscript PDF file)?

Reviewer #1: Yes

Reviewer #2: Yes

Reviewer #3: Yes

4. Is the manuscript presented in an intelligible fashion and written in standard English?

Reviewer #1: No

Reviewer #2: Yes

Reviewer #3: No

5. Review Comments to the Author

Reviewer #1: This is a good paper. However, it needs more work, and I recommended the authors to do the following;

1. The whole discussion needs to be written and synthesized. I think there are so many repetitive details, spelling, and grammatical errors.

2. Results need to be presented succinctly and clearly. There is so much repetition also noted

3. The authors also need to maintain a good academic tone, use academic words, and maintain coherence and clarity.

4. The authors need to check that all abbreviations are well used and defined.

5. Authors need to recheck all punctuations and correct prepositions as stated in the review draft.

Reviewer #2: Title of the paper: Number of antenatal care visits and associated factors among reproductive age women in Sub-Saharan Africa using recent demographic and health survey data from 2008-2019: a multilevel negative binomial regression model

Reviewer comments:

Specific comments:

Lines 105-114. The authors state the following:

The source population was all reproductive-age women who gave birth five years preceding each respective survey in sub-Saharan Africa, whereas those in the selected Enumeration Areas (EAs) were the study population. The sample size was determined from the kids to recode file “KR file” from the standard DHS dataset of Sub-Saharan African countries with at least one survey from 2008 to 2019. DHS uses a two-stage stratified cluster sampling technique. In the first stage, a sample of EAs was selected independently from each stratum with proportional allocation stratified by residence (urban & rural). In the second stage, from the selected EAs, households were taken by systematic sampling technique (23).

Comment: This narrative only tells us how the DHS selected the samples from within each country. But does not tell us how this process weights the sample across the 35 countries in the five identified geo-regions of Africa. This is necessary to establish how the computations of the merged sample was undertaken.

Lines 114-116: The authors report the weighted sample size. This is not appropriate while reporting. The analysis uses the weighted sample. However, while reporting the actual number of women included needs to be reported. But in lines 195-196 in the results section the authors mention ” A total of 256,425 reproductive-age women who gave birth within five years preceding the survey were included’. The weighted sample is also listed as 256, 425. The authors need to clarify this.

Lines 150-151. Sample weighting was done for each country before appending it to one

151 dataset.

This is not an adequate way of weighing to obtain estimates for the DHS data sets of Africa. The authors need to explain or substantiate the process of using sub-national weights which will yield appropriate estimates for the nation across its strata are also equally applicable for the nation as a whole when it is merged into the collective African data set. This has not been described and it probably is not done. That renders the whole analysis invalid.

It is likely that the model originally envisaged, that is the Poisson regression model because of this particular factor which would have demonstrated high variability because of the lower variation within a country and higher across the country thereby yielding a high sample variance (about two times that of the mean) thereby resulting in the choice of the negative binominal regression model. The authors seem to have overlooked this possibility.

Lines 193-195: The authors say: Variance inflation factor (VIF) was assessed to check for multi-linearity and the mean VIF for the fitted model was 1.86. Finally, a Model comparison was done using the deviance test, and the model with the lowest deviance was selected as the best-fitted model.

Comment: The VIF shown is only for the last model that the authors present. However, each of the regressions in the exercise carries considerable collinearity that the authors do not check there by rendering the exercise inadequate.

Overall comment:

The analysis is not appropriate as it has not described the process of weighing used to merge the African data set with appropriate weights. Further, the selection of variables within each model has not been justified using the VIF- where several variables could be conceptually associated, particularly maternal education, wealth index and media usage in model 2.

Given these limitations in analysis, and the need for computing weights for the continental analysis envisaged, this paper does not yield valid results for interpretation. I will not be able to recommend it for publication in its present form.

Reviewer #3: The analysis in the paper has used excellent analytical tool with proper justification of the method but not specified the research questions and theoretical conceptual model. As a result, they may not be clear why to analyze ANC service utilization as a count variable and what may be it' programmatic implications.

Analysis using multilevel Negative Binomial distribution has not incorporated variation in model paraments under fixed effect across different social groups and also the implicit variability across sub-Saharan African countries have not been explored adequately. Thus, the paper may be accepted for publication is the authors revise it in the context of these 

Pls download special comments in the manuscript as well attached in the system. 

6. PLOS authors have the option to publish the peer review history of their article (what does this mean?). If published, this will include your full peer review and any attached files.

**Do you want your identity to be public for this peer review?** For information about this choice, including consent withdrawal, please see our Privacy Policy.

Reviewer #1: **Yes: **Lydia Sandrah Kaforau

Reviewer #2: No

Reviewer #3: No

---

## [Editor Report · Decision Letter 1]

24 Oct 2022

PGPH-D-22-00897R1

Number of antenatal care visits and associated factors among reproductive age women in Sub-Saharan Africa using recent demographic and health survey data from 2008-2019: a multilevel negative binomial regression model

Dear Dr. Zemene,

Thank you for submitting your manuscript to PLOS Global Public Health. After careful consideration, we feel that it has merit but does not fully meet PLOS Global Public Health’s publication criteria as it currently stands. Therefore, we invite you to submit a revised version of the manuscript that addresses the points raised during the review process.

We look forward to receiving your revised manuscript.

Kind regards,

Jitendra Kumar Singh, PhD

Academic Editor

Journal Requirements:

When submitting your revision, we need you to address these additional requirements.                                         . Please ensure that your manuscript meets PLOS GLOBAL PUBLIC HEALTH's style requirements, including those for file naming. 

4. PLOS GLOBAL PUBLIC HEALTH does not copyedit accepted manuscripts, so the language in submitted articles must be clear, correct, and unambiguous. We recommend that authors seek independent editorial help before submitting a revision. These services can be found on the web using search terms like “scientific editing service” or “manuscript editing service.”

Additional Editor Comments (if provided):

Sufficient details were not provided on the sample size calculation and sampling procedure to enable replication of your study. Please provide sufficient detail to allow reviewers to evaluate whether the conclusions of the study presented are generalizable to the wider population.

What items were considered when calculation wealth index, pls list-out. 

Why Negative binomial distribution used? Pls provide sufficient details on selecting Negative binomial distribution with respect of this manuscript as authors discussed the Poisson regression model as well. What is the basis of selecting a p-value ≤ 0.2 as candidates for the final model. 

Reviewers' comments:

Reviewer-1

This is a good paper. However, it needs more work, and I recommended the authors to do the following;

1. The whole discussion needs to be written and synthesized. I think there are so many repetitive details, spelling, and grammatical errors.

2. Results need to be presented succinctly and clearly. There is so much repetition also noted

3. The authors also need to maintain a good academic tone, use academic words, and maintain coherence and clarity.

4. The authors need to check that all abbreviations are well used and defined.

5. Authors need to recheck all punctuations and correct prepositions as stated in the review draft.

Also see comments in review pan in Attached file in the system.

Reviewer 2

Specific comments:

Lines 105-114. The authors state the following:

The source population was all reproductive-age women who gave birth five years preceding each respective survey in sub-Saharan Africa, whereas those in the selected Enumeration Areas (EAs) were the study population. The sample size was determined from the kids to recode file “KR file” from the standard DHS dataset of Sub-Saharan African countries with at least one survey from 2008 to 2019. DHS uses a two-stage stratified cluster sampling technique. In the first stage, a sample of EAs was selected independently from each stratum with proportional allocation stratified by residence (urban & rural). In the second stage, from the selected EAs, households were taken by systematic sampling technique (23).

Comment: This narrative only tells us how the DHS selected the samples from within each country. But does not tell us how this process weights the sample across the 35 countries in the five identified geo-regions of Africa. This is necessary to establish how the computations of the merged sample was undertaken.

Lines 114-116: The authors report the weighted sample size. This is not appropriate while reporting. The analysis uses the weighted sample. However, while reporting the actual number of women included needs to be reported. But in lines 195-196 in the results section the authors mention ” A total of 256,425 reproductive-age women who gave birth within five years preceding the survey were included’. The weighted sample is also listed as 256, 425. The authors need to clarify this.

Lines 150-151. Sample weighting was done for each country before appending it to one

151 dataset.

This is not an adequate way of weighing to obtain estimates for the DHS data sets of Africa. The authors need to explain or substantiate the process of using sub-national weights which will yield appropriate estimates for the nation across its strata are also equally applicable for the nation as a whole when it is merged into the collective African data set. This has not been described and it probably is not done. That renders the whole analysis invalid.

It is likely that the model originally envisaged, that is the Poisson regression model because of this particular factor which would have demonstrated high variability because of the lower variation within a country and higher across the country thereby yielding a high sample variance (about two times that of the mean) thereby resulting in the choice of the negative binominal regression model. The authors seem to have overlooked this possibility.

Lines 193-195: The authors say: Variance inflation factor (VIF) was assessed to check for multi-linearity and the mean VIF for the fitted model was 1.86. Finally, a Model comparison was done using the deviance test, and the model with the lowest deviance was selected as the best-fitted model.

Comment: The VIF shown is only for the last model that the authors present. However, each of the regressions in the exercise carries considerable collinearity that the authors do not check there by rendering the exercise inadequate.

Overall comment:

The analysis is not appropriate as it has not described the process of weighing used to merge the African data set with appropriate weights. Further, the selection of variables within each model has not been justified using the VIF- where several variables could be conceptually associated, particularly maternal education, wealth index and media usage in model 2.

Given these limitations in analysis, and the need for computing weights for the continental analysis envisaged, this paper does not yield valid results for interpretation.

I will not be able to recommend it for publication in its present form.

Reviewer 3

The analysis in the paper has used excellent analytical tool with proper justification of the method but not specified the research questions and theoretical conceptual model. As a result, they may not be clear why to analyze ANC service utilization as a count variable and what may be it' programmatic implications.

Analysis using multilevel Negative Binomial distribution has not incorporated variation in model paraments under fixed effect across different social groups and also the implicit variability across sub-Saharan African countries have not been explored adequately. Thus, the paper may be accepted for publication is the authors revise it in the context of these

---

## [Editor Report · Decision Letter 2]

28 Nov 2022

Number of antenatal care visits and associated factors among reproductive age women in Sub-Saharan Africa using recent demographic and health survey data from 2008-2019: a multilevel negative binomial regression model

PGPH-D-22-00897R2

Dear Mr. Zemene,

We are pleased to inform you that your manuscript 'Number of antenatal care visits and associated factors among reproductive age women in Sub-Saharan Africa using recent demographic and health survey data from 2008-2019: a multilevel negative binomial regression model' has been provisionally accepted for publication in PLOS Global Public Health.

Best regards,

Jitendra Kumar Singh, PhD

Academic Editor
